# Improvement of pBRDF Model for Target Surface Based on Diffraction and Transmission Effects

Qiang Fu [1,2,*], Xuanwei Liu [1,2], Di Yang [3], Juntong Zhan [1,2], Qing Liu [3], Su Zhang [1,2], Fang Wang [3], Jin Duan [1], Yingchao Li [2] and Huilin Jiang [2]

1 College of Opto-Electronic Engineering, Changchun University of Science and Technology, Changchun 130022, China; 2021100272@mails.cust.edu.cn (X.L.); zhanjuntong@cust.edu.cn (J.Z.); zhangsu@cust.edu.cn (S.Z.); duanjin@cust.edu.cn (J.D.)
2 Space Opto-Electronics Technology Institute, Changchun University of Science and Technology, Changchun 130022, China; liyingchao@cust.edu.cn (Y.L.); hljiang@cust.edu.cn (H.J.)
3 Beijing Institute of Tracking and Telecommunication Technology, Beijing 100094, China; 2021100256@mails.cust.edu.cn (D.Y.); 2022100408@mail.cust.edu.cn (Q.L.); 2022100279@mails.cust.edu.cn (F.W.)
* Correspondence: fuqiang@cust.edu.cn

**Abstract:** The polarised Bidirectional Reflectance Distribution Function (pBRDF) model relates the properties of target materials to the polarisation information of the incident and reflected light. The Priest–Germer (P-G) model was the first strictly pBRDF model to be officially released; however, some shortcomings remain. In this study, we first analyse the assumption framework of the P-G model, analyse the assumption framework to determine the imperfections in the framework, supplement the boundary conditions of the model for diffraction and transmission effects, and propose and construct a polarised pBTDF model based on the existing P-G model and parameter inversion; the output results of the model are compared with the experimental data through simulation. The results show that the intensity relative error and Degree of Linear Polarisation relative error of the target can be reduced by more than 40%, using the improved model, proving its accuracy and precision.

**Keywords:** diffraction effect; transmission effect; parameter inversion; P-G model; pBTDF model

## 1. Introduction

The asymmetry of the vibration direction of light with respect to the propagation direction is called polarisation and is the most obvious marker that distinguishes transverse waves from other longitudinal waves. During the propagation of light, different targets exhibit different polarisation properties according to their different properties [1–3].

The Bidirectional Reflectance Distribution Function (BRDF) describes the distribution of incident light in each of the outgoing directions after reflection from a given surface, and a great deal of BRDF research has been conducted worldwide since the 1980s, with significant progress being made [4–6]. Early studies focused on model simulation software for the polarisation properties [7]. With the accumulation of data and advancement of knowledge, researchers have begun to study incident light in depth, the mechanism of reflected light generation, and the inversion of target parameters [8,9].

To be able to reflect the polarisation characteristics of the reflection from the surface of the object, the polarised BRDF (pBRDF) model is introduced, which is a more general form of BRDF, with expressions similar to BRDF; however, the scalar values in the expressions of BRDF are changed to vector values. Since the year 2000, scholars have gradually extended and improved the pBRDF model [10–12]. However, there is still a lack of studies to model typical coating materials based on diffraction and transmission effects as the theoretical basis. Priest of the US Naval Laboratory successfully incorporated the traditional Torrance–Sparrow model (T-S model) in the same year after coupling it with Mueller matrix polarisation [13]. In 2000, Priest and Germer used the T-S model as the base model and formally published the first strict pBRDF model, the Priest–Germer (P-G) model [14].

The traditional P-G model still has incomplete parts: it is based on the combination of specular micro-elements, which need to have a scale much larger than the wavelength; this means that when the scale of the specular surface is smaller or comparable to the wavelength, the phase relationship between photons will change randomly because of the diffraction effect, and this random change of phase will lead to the reduction of Degree of Linear Polarisation (DOLP). The transmission effect is a phenomenon that causes the light to rotate relative to the original specular reflection direction because of the presence of either a curved or convex target surface shape. For the diffused-reflection type photons, each bump on the target surface can be regarded as a combination of numerous mirrors and a rearrangement of mirrors according to the transmission effect. Both effects cause changes in the polarisation characteristics of the target surface; thus, in this study, we aimed to improve the traditional P-G model for diffraction and transmission effects and propose a new model with higher accuracy and better applicability.

The nonlinear least squares method is used to invert the parameters of the target to find the best combination of parameters that minimises the standard deviation between the experimental measurement results and the model simulation results. The best parameters obtained from the inversion are then substituted into the model, and the fitting degree of the measured and simulated data of the traditional P-G model and the improved model are compared. The results show that the relative error of the improved intensity model can be reduced by more than 42.8% compared with the P-G model, and the relative error of the improved DOLP model can be reduced by more than 16.83% compared with the P-G model, thus proving that the improved model has higher accuracy. This can provide a theoretical support for analysing the polarisation characteristics of a target surface.

## 2. Materials and Methods

### 2.1. Overview of P-G model

The traditional P-G model is as follows:

$$
\begin{aligned}
f_{ij}(\theta_i, \theta_r, \varphi_i, \varphi_r, \lambda) &= f_{spec} + f_{vol}(\theta_i, \theta_r) \\
&= \frac{m_{ij} f_{SO}(\theta, \beta, \tau, \Omega) P(\theta, \sigma, B_n)}{4 \cos \theta_i \cos \theta_r} + \rho_d + \frac{2\rho_v}{\cos \theta_i + \cos \theta_r},
\end{aligned}
\tag{1}
$$

where $f_{ij}$ is any element value in the BRDF matrix of the P-G model polarisation; $f_{vol}$ is the scattering component, reflecting the energy characteristics of the P-G model, which can be further divided into the diffuse reflection component, $\rho_d$, and bulk scattering component, $\rho_v$; $f_{spec}$ is the specular reflection component, reflecting the polarisation characteristics of the P-G model, and can be derived from the micro-plane element theory; $m_{ij}$ is the element of the Mueller matrix; $P(\theta, \sigma, B_n)$ is the probability distribution function of the micro-plane element normal direction and has Gaussian and Cauchy distributions; $\theta_i$ and $\varphi_i$ are the zenith and azimuth angles in the incident direction, respectively; $\theta_r$ and $\varphi_r$ represent the zenith and azimuth angles in the reflected direction, respectively; $\lambda$ is the wavelength; $f_{SO}$ is the shading and occlusion function; $\theta$ is the angle between the specular micro-element normal and material surface normal; $\beta$ denotes the angle between the incident and outgoing rays; $\sigma$ is the object surface roughness constant; and $B_n$ is the deviation of the micro-element normal direction from the parameter of the size of the mean normal direction.

The Müller matrix is a function derived from Fresnel's formula [15], and the reflected Müller matrix is as follows:

$$
M_r = \begin{bmatrix} m_{00} & m_{10} & m_{20} & m_{30} \\ m_{10} & m_{11} & m_{21} & m_{31} \\ m_{20} & m_{21} & m_{22} & m_{32} \\ m_{30} & m_{31} & m_{23} & m_{33} \end{bmatrix} = R \begin{bmatrix} 1 & \cos 2\psi & 0 & 0 \\ \cos 2\psi & 1 & 0 & 0 \\ 0 & 0 & \sin 2\psi \cos \Delta & \sin 2\psi \sin \Delta \\ 0 & 0 & -\sin 2\psi \sin \Delta & \sin 2\psi \cos \Delta \end{bmatrix},
\tag{2}
$$

where $R$ is the reflectivity, $\psi$ is the tangent angle formed by the ratio of the amplitudes of $p$ and $s$ waves, and $\Delta$ is the phase difference between the $p$ and $s$ waves.

The shading and blocking functions reflect the expression of the incident light-shading and outgoing light-blocking law, owing to the undulation of the rough surface of the material, which is expressed as follows [16]:

$$SO(\theta, \beta, \tau, \Omega) = \frac{1 + \frac{\theta}{\Omega} e^{-2\beta/\tau}}{1 + \frac{\theta}{\Omega}}, \tag{3}$$

where $\tau$ and $\Omega$ are parameters characterising the shadowing and shading effects on rough surfaces; the values obtained are different depending on the material and can be obtained by measurement; and $\theta$ and $\beta$ are variables, and the expressions for both are shown below.

$$\cos(2\beta) = \cos\theta_i \cos\theta_r + \sin\theta_i \sin\theta_r \cos(\varphi_r - \varphi_i), \tag{4}$$

$$\cos(\theta) = \frac{\cos\theta_i + \cos\theta_r}{2\cos\beta}, \tag{5}$$

where $\beta$ denotes the angle between the incident and outgoing rays, and $\theta$ is the angle between the normal of the specular micro-element and that of the material surface.

### 2.2. P-G Model Error Analysis

In this study, the polarised two-way reflection distributions of light green, dark green, and earthy yellow lacquers on an aluminium plate substrate were evaluated. Figure 1 shows a comparison of the test data (represented by star points) for the light green, dark green, and earth yellow lacquer layers on an aluminium plate substrate and the simulation data (represented by connecting lines) obtained based on the inversion of the P-G model parameters.

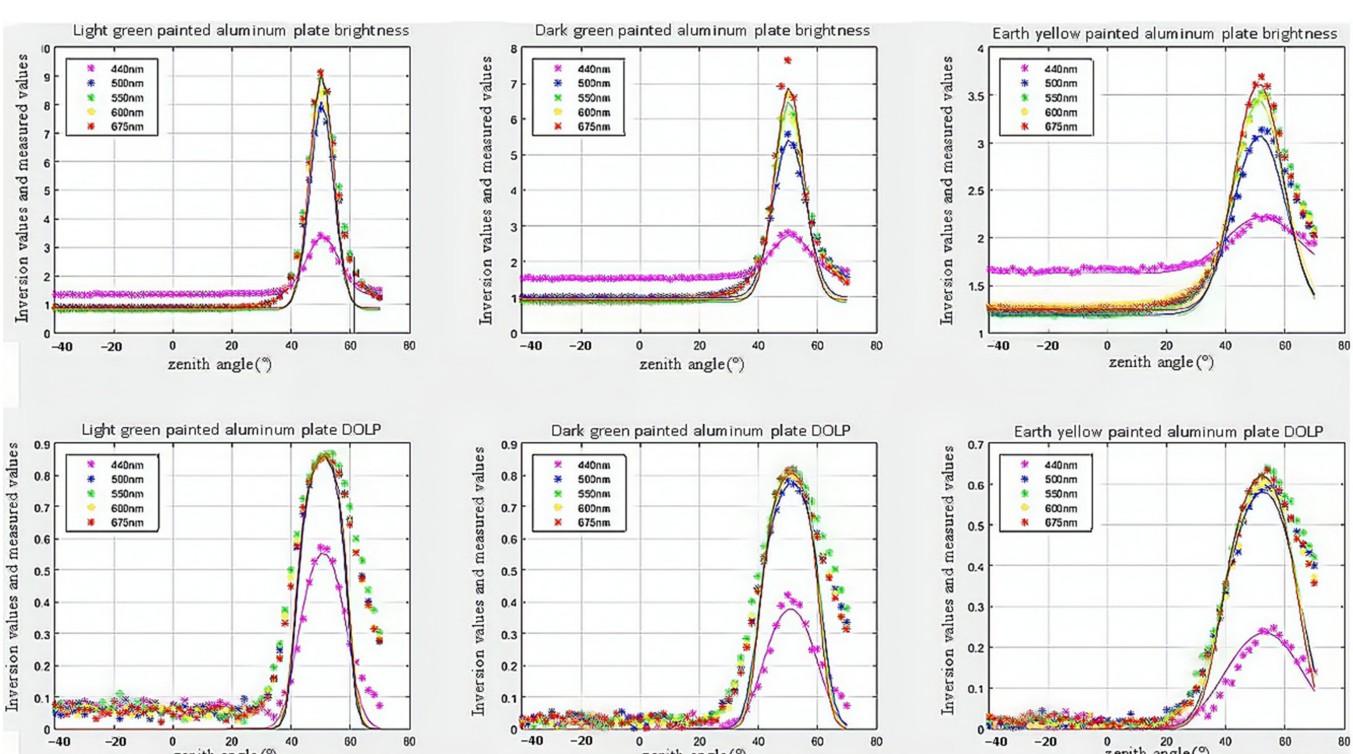

**Figure 1.** Comparison of test data for light green, dark green, and earthy yellow paint layers with an aluminium plate as the substrate and simulation data obtained based on the inversion of the Priest–Germer (P-G) model parameters.

As can be seen in Figure 1, there are certain deviations between the P-G model data and the measured data for both intensity and linear polarisation, and the modelling accuracy of

intensity has a greater impact on the modelling accuracy of polarisation, which needs to be improved by new methods to improve the modelling accuracy.

### 2.3. The Importance of Model Improvement

The P-G model and its derivative models are usually applicable to materials with a particle size that is larger than the wavelength, such as metals, plastics, and glass. However, for typical coating materials, the particle size is usually between 10 and 100 μm, which is much smaller than the wavelength, and the diffraction effect will occur at this time, so that the propagation path of light will be bent after encountering the object particles in the process of propagation, resulting in random changes in the phase relationship between particles and photons. At this time, it is difficult to describe this part with the P-G model, resulting in reduced model accuracy.

The usual coating materials, such as paint, have low reflectivity and high transmittance. For this kind of material which contains radiation start and high transmittance, the distribution function is a combination of the distribution function of external incident light and a distribution function of positive incident light from inside, and it is difficult for the traditional pBRDF model to accurately simulate this kind of effect.

Coated materials are widely used in various fields, such as spacecraft, military weaponry, medical devices, etc. Therefore, modelling diffraction and transmission effects can improve the fitting and prediction accuracy for such targets.

For most paint targets, such as paints, the surface roughness constants are usually between 0.2 and 0.8, much greater than materials such as metals and glass. Because it is a mixture formed by lipids, pigments, etc., the surface layer of the material is also a mixture of states, which, at the macro level, is rougher. Moreover, because of the process of painting, the paint film is not uniform, there are particles in the environment, and other factors cannot be avoided. Therefore, the surface roughness of the coating material is larger, and its surface undulation is much larger than the wavelength.

### 2.4. Model Improvement Based on Diffraction Effects

The P-G model is based on a combination of mirrored micro-elements, which need to have a scale much larger than the wavelength, meaning that when the scale of the mirrors is smaller or comparable to the wavelength, the presence of the diffraction effect leads to a random change in the phase relationship between the photons, and this random change in phase leads to a decrease in polarisation.

First, we examined the phase coherence of the light field in the upper hemisphere owing to undulating changes in the surface of the face pattern. Without considering the phase difference because of the physical properties of the material, the phase difference because of the surface pattern is shown in Figure 2; because the surface of the material has homogeneous characteristics, the three-dimensional case can be replaced by a two-dimensional one.

At any point in the upper hemisphere space is the effect of a phase-coherent superposition of the light intensity of the undulating surface of the material. Without considering the effect of the phase angle because of the angle of the incident and reflected light, the distribution of the intensity in the upper hemisphere space can be seen as a superposition effect per micro-element [17]; at an observation point $(x_{rj}, y_{rj})$, the phase superposition of the intensity is as follows:

$$E = \sum_{i=1}^{i=n} \left[ (x_i - x_{rj})^2 + (y_i - y_{rj})^2 \right]^{0.5}. \tag{6}$$

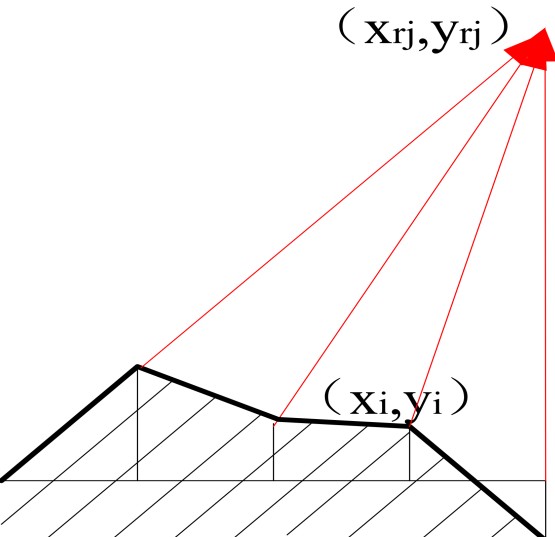

**Figure 2.** Distance between an undulating surface and any point in space.

In the pBRDF model, the expression of each effect in the model is the product of the coefficients of that part and the intensity distribution. When the diffraction effect occurs, the intensity distribution on space, that is, the light field distribution (E), is superimposed by each micro-element because the phase relationship between photons changes randomly. Equation (6) corresponds to the light field superposition pattern at a point in space, that is, the intensity distribution. The intensity distribution of the diffraction effect on space can be derived from Equation (6), and then the model component based on the diffraction effect can be deduced.

When the wavelength is 500 nm, the sampling interval is 200 nm, and the maximum peak and valley values of the surface undulation are 50 nm, 500 nm, and 2 μm; the intensity distribution on the near surface is shown in Figure 3.

As shown in Figure 3, there is an undulation of the light intensity in the vicinity of the material. As the distance from the surface increases, the effect of the phase difference caused by the surface undulation gradually decreases, and it is mainly the distance that plays a role, not the surface undulation. As the surface undulation gradually increases, the area of light intensity undulation becomes longer, but, in general, the undulation of light intensity tends to be homogeneous once the distance exceeds a certain value.

For coated materials, the surface undulation is much greater than the wavelength. The characterisation of the depolarisation properties is only angle dependent if the surface undulation is similar to the wavelength; therefore, the intensity distribution of the diffraction part, $P_b$, can be replaced by a constant when modelling.

As shown in Figure 3, as the surface undulation of the target increases, $P_b$ gradually converges to 1 because, at sufficiently large surface undulations, the polarisation state of the incident light can be considered to be random with the direction of interaction between the photons and particles within the material; hence, the polarisation state of the incident light is randomly distributed. However, the diffracted light is equally likely to be distributed in all directions throughout space. Therefore, for materials such as paint, for which the surface undulation is much greater than the wavelength, the diffraction effect can be considered a result of having the same intensity in all directions. Thus, $P_b$ can be defined as follows:

$$P_b = 1 \times M_{ij}{}^b, \tag{7}$$

where $M_{ij}{}^b$ is the Mueller matrix element of the diffraction.

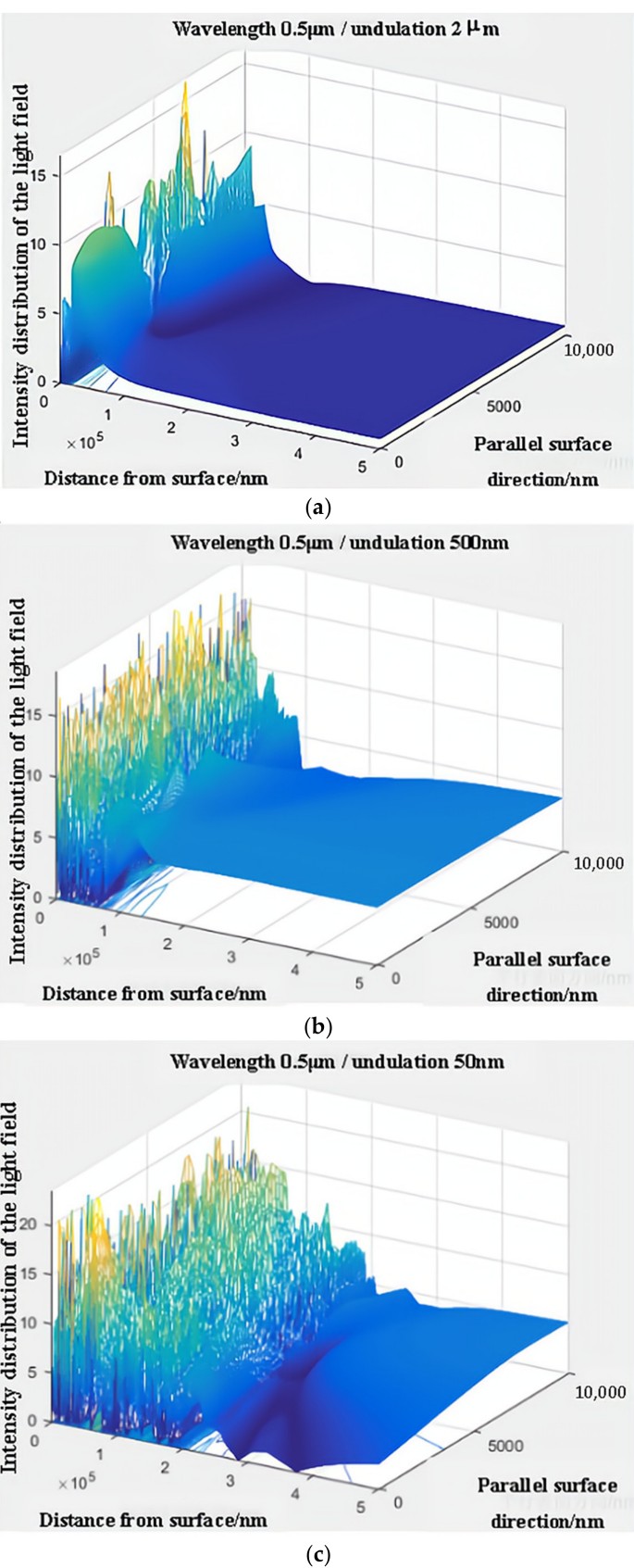

**Figure 3.** Strength distribution on materials. (**a**) Undulation of 2 μm. (**b**) Undulation of 500 nm. (**c**) Undulation of 50 nm.

### 2.5. Model Improvement Based on Transmission Effects

It is assumed that, for a specular surface in a specular micro-element assembly, the atoms and molecules within it are undirected and unbiased sources of light, as shown in Figure 4a. At the surface of the material, the scattered light from all directions is equal at each point, as shown in Figure 4b. It may be useful to bend the face shape at each point so that the incident light is parallel with respect to the surface normal to the mirror; that is, it is assumed that the mirror is composed of an infinite number of surfaces with circular projections and that there is positive incident light inside, as shown in Figure 4c. Here, there is a rotation of the transmitted light with respect to the original mirror, with the angle of rotation being equal to the angle of rotation of the mirror. The mirrors were rearranged, considering each projection as a combination of an infinite number of mirrors so that, for the mirrors, the effect of the radiation transmitted from the interior corresponds to the scattering of a set of parallel lights after they have been incident on a convex lens, as shown in Figure 4d.

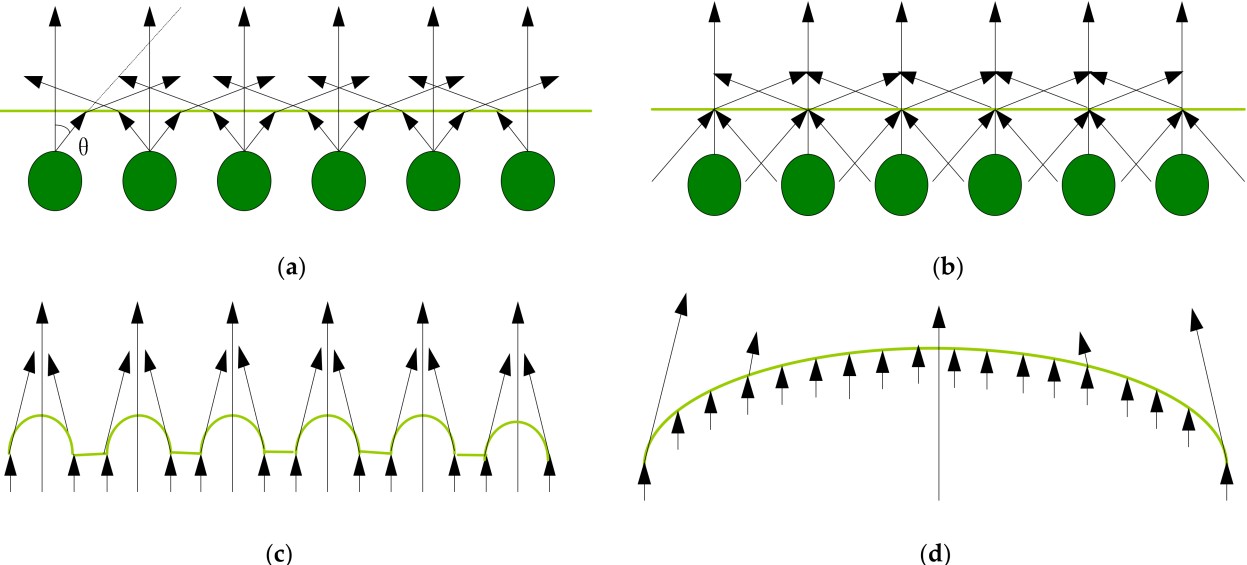

(**a**)

(**b**)

(**c**)

(**d**)

**Figure 4.** Illustration of light transmission from a specular micro-element. (**a**) Scattering of atomic molecules. (**b**) Surface transmitted light. (**c**) Single specular micro-element transformation. (**d**) Micrometric statistical transformations.

Based on the above hypothetical conclusions, the pBRDF of transmitted light can be analysed using the reflection and transmission of the specular micro-element as the primitive. From the Fresnel metric [16], the polarisation state of the light reflected from a specular micro-element can be expressed as follows:

$$\begin{pmatrix} I_r \\ Q_r \\ U_r \\ V_r \end{pmatrix} = M_r \begin{pmatrix} I_o \\ Q_o \\ U_o \\ V_o \end{pmatrix},$$

(8)

where $I$, $Q$, $U$, and $V$ denote the Stokes vectors of the incident and reflected light; the subscript o denotes the incident light; $r$ denotes the reflected light; and $M_r$ denotes the Mueller matrix of reflection.

In the case of transmitted light, similar to the reflected light, the polarisation state of the light transmitted by a specular micro-element can be expressed as follows:

$$\begin{pmatrix} I_r \\ Q_r \\ U_r \\ V_r \end{pmatrix} = M_t \begin{pmatrix} I_t \\ Q_t \\ U_t \\ V_t \end{pmatrix}, \tag{9}$$

where the subscript $r$ of the observed Stokes vector is also used to represent the transmitted light, as the angle of observation is the same for the transmitted and reflected light; and $M_t$ represents the Mueller matrix of transmission.

Referring to the general form of the P-G model for specular transmission, we propose the pBTDF model, which can be expressed by the following equation:

$$f_{t-spec} = \frac{M_t(\beta_t, n, k) f_{t-SO}(\theta_t, \beta_t, \tau, \Omega) P_t(\theta_t, \sigma, B_n)}{4 \cos \theta_r}. \tag{10}$$

Here, the subscript $t$ denotes transmission (radiation), $I$ denotes incidence, the incidence angle of $0°$, and $r$ denotes refraction (in the same direction as a reflection in pBRDF). The equivalent incidence angle, $\beta_t$, and mean normal deviation angle, $\theta_t$, in $f_{t-SO}$ and $P_t$ are expressed as follows:

$$\cos(2\beta_t) = \cos \theta_i \cos \theta_r + \sin \theta_i \sin \theta_r \cos(\varphi_r - \varphi_i) = \cos \theta_r, \tag{11}$$

$$\cos(\theta_t) = \frac{\cos \theta_i + \cos \theta_r}{2 \cos \beta_t} = \cos(\frac{\theta_r}{2}). \tag{12}$$

Namely,

$$\theta_t = \beta_t = \frac{\theta_r}{2}. \tag{13}$$

With reference to the general form of the shading functions in the P-G model and the probability distribution function of the surface normal, $f_{t-SO}$ and $P_t$ can be expressed as follows:

$$f_{t-SO} = \frac{1 + \frac{\theta_t}{\Omega} e^{-2\beta_t/\tau}}{1 + \frac{\theta_t}{\Omega}} = \frac{1 + \frac{\theta_r}{2\Omega} e^{-\theta_r/\tau}}{1 + \frac{\theta_r}{2\Omega}}, \tag{14}$$

$$P_{t-G} = \frac{B \exp(-\frac{\tan(\theta_t)}{2\sigma^2})}{2\pi\sigma^2 \cos^3(\theta_t)} = \frac{B \exp(-\frac{\tan(\frac{\theta_r}{2})}{2\sigma^2})}{2\pi\sigma^2 \cos^3(\frac{\theta_r}{2})}. \tag{15}$$

Here, the subscript $G$ denotes a Gaussian distribution; the parameters $\Omega$, $\tau$, and $B$ are constants with different values depending on the material; and $\sigma$ is a parameter related to roughness.

We analysed the effects of the transmitted light on pBRDF and pBTDF in terms of both intensity and polarisation.

When the angle of incidence of light is $50°$, the light green lacquer coating, for example, has a reflectance of 0.0790 and a complex refractive index of 1.39 + 0.3371i [18]. Since the effect of transmitted light is considered in this study, the non-specular reflective part is not brought into the model for calculation, and the absorption conversion efficiency of the light green lacquer aluminium plate on the incident light is also not considered; the simulation calculation results are shown in Figure 5. In Figure 5, the top graph shows the reflected brightness distribution curve, and the bottom graph shows the polarisation distribution curve. The solid blue line represents the reflection, the dashed red line represents the transmission, and the dotted line represents the total reflection and transmission.

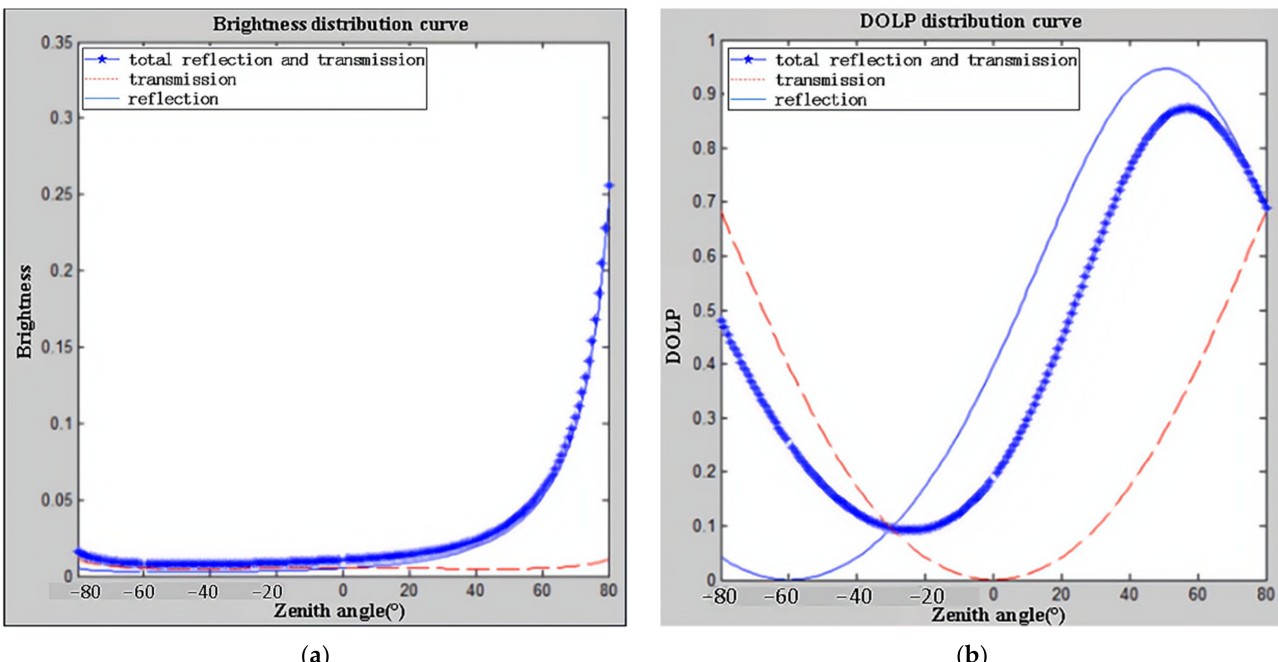

**Figure 5.** Brightness and polarisation distribution curves for light green painted aluminium plates at an incidence angle of 50°. (**a**) Brightness distribution of green lacquered panels. (**b**) Polarisation distribution of green lacquered panels.

As can be seen from Figure 5a, owing to the low reflectivity of the green paint, the difference between the brightness of the reflected and transmitted light is smaller when the phase angle is small; as the phase angle increases, the ratio of reflected to transmitted light gradually increases, implying that, at small phase angles, the transmitted light has a greater influence on the overall polarisation characteristics. As can be seen in Figure 5b, at observation angles less than −40°, the actual DOLP is closer to the DOLP of the transmitted light; however, at observation angles greater than 0°, the DOLP is closer to the DOLP of the reflected light, indicating that the ratio of reflected to transmitted light and the DOLP of the reflected to transmitted light together determine the global DOLP. In addition, the polarisation of the transmitted light also leads to an increase in the DOLP for nonpositive incident light, which is not zero throughout the observed hemispheric space, even when observed at the angle of incidence.

The combined analysis of the transmission effect leads to the following conclusion: modelling of the transmission effect in the visible wavelength band is essentially the modelling of partially, nonspecularly reflected light and can reflect, to some extent, the reasons for the shaping of the nonspecularly reflected component.

*2.6. Model Improvement*

Combining these two aspects, we propose the following scenario: the paint, at the macroscopic level, can be considered a homogeneous mixture with a structural scale much larger than the wavelength so that the intensity distribution of its diffractive part tends to be homogeneous. The transmitted part has a high weight in the visible range, and together with the reflected part, it determines global polarisation. Therefore, the improved model consisted of two main components: a specular-reflecting component and a transmitting component.

2.6.1. Specular Reflection

First, the specular micro-elements should have a compound-symmetric distribution when studying single-particle scattering. The Cauchy distribution was not symmetrical; thus, it was not considered. If we use a Gaussian distribution, there is a problem of low

peaks and troughs. The approximate description of the particle scattering phase function is referred to as the Hermite–Gaussian (H-G) phase function [19], which is used to describe the asymmetry of the light intensity in the forward–backwards intensity distribution. Therefore, this H-G phase function is used to describe the specular micro-element normal distribution of the material surface, whereupon, for the specular reflection,

$$P_r = \frac{m_{ij} f_{SO}(\theta, \beta, \tau, \Omega) P_{RHG}(\cos\theta)}{4\cos\theta_i \cos\theta_r}, \tag{16}$$

where $P_{RGH}(\cos\theta)$ is the surface normal probability distribution of the specular micro-element, which is expressed as

$$P_{RHG}(\cos\theta) = \frac{1 - g_r^2}{\left(1 + g_r^2 - 2g_r\cos\theta\right)^{1.5}}. \tag{17}$$

Here, $g_r$ is the asymmetry factor.

### 2.6.2. Transmission

Having mentioned the general form of the probability distribution function for the transmission component in the previous section, the phase function for the transmission component, after describing it in terms of the H-G phase function, is as follows:

$$P_t = \frac{M_{ij}{}^t f_{t-SO}(\theta_t, \beta_t, \tau, \Omega) P_{THG}(\cos\theta_t)}{4\cos\theta_r}. \tag{18}$$

Here, $P_{THG}$ is the surface normal probability distribution of the transmission effect as follows:

$$P_{THG}(\cos\theta_t) = \frac{1 - g_t^2}{\left(1 + g_t^2 - 2g_t\cos\theta_t\right)^{1.5}}, \tag{19}$$

where $g_t$ is the asymmetry factor in transmission.

### 2.6.3. Model Synthesis

Combining the above studies, we need to model not only the reflected light component but also the transmitted and diffracted light components based on the P-G model; therefore, based on the P-G model, our proposed new pBTDF model is described in terms of both intensity and DOLP, and the improved model is as follows.

For the intensity section:

The improved model based on transmission and diffraction effects mainly consists of three parts, namely specular reflection, transmission, and diffraction, which are treated separately in the study to obtain the expressions for the intensity part of the pBTDF model as follows:

$$f_{pBTDF} = \rho_r \cdot P_r + \rho_t \cdot P_t + \rho_b \cdot P_b, \tag{20}$$

where $\rho_r$, $\rho_t$, and $\rho_b$ denote the coefficients of the specular reflection, transmission, and diffraction components, respectively.

By substituting Equations (7), (16), and (18) into Equation (20), respectively, we obtain the following:

$$\begin{aligned} f_{pBTDF} = \rho_r \times \frac{M_{ij}{}^r f_{SO}(\theta, \beta, \tau, \Omega) P_{RHG}(\cos\theta)}{4\cos\theta_i \cos\theta_r} \\ + \rho_t \times \frac{M_{ij}{}^t f_{t-SO}(\theta_t, \beta_t, \tau, \Omega) P_{THG}(\cos\theta_t)}{4\cos\theta_r} + \frac{\rho_b \times M_{ij}{}^b}{\cos\theta_i \cos\theta_r} \end{aligned} \tag{21}$$

For the DOLP section:

Because the Mueller matrix is a 4 × 4 matrix, but the circular polarisation component is small and usually neglected in the study of polarisation properties, the incident light and pBTDF

matrix can be downscaled after neglecting the circular polarisation component. Therefore, Equation (21) can be converted into a $3 \times 3$ matrix form with the following expression:

$$f_{pBTDF} = \begin{bmatrix} f_{00} & f_{01} & f_{02} \\ f_{10} & f_{11} & f_{12} \\ f_{20} & f_{21} & f_{22} \end{bmatrix} = \begin{bmatrix} \rho_r P_{r00} + \rho_t P_t + \rho_b P_b & \rho_r P_{r01} & \rho_r P_{r02} \\ \rho_r P_{r10} & \rho_r P_{r11} & \rho_r P_{r12} \\ \rho_t P_{t20} & \rho_t P_{t21} & \rho_t P_{t22} \end{bmatrix}. \tag{22}$$

In the case where the incident light is $0°$ linearly polarised, the incident light is expressed in terms of the Stokes vector, and the reflected light Stokes vector is obtained with the following expression:

$$\begin{aligned} \boldsymbol{L}_r &= \begin{bmatrix} L_0 \\ L_1 \\ L_2 \end{bmatrix} = \begin{bmatrix} I \\ Q \\ U \end{bmatrix} = \begin{bmatrix} E_0 \\ E_1 \\ E_2 \end{bmatrix} = \begin{bmatrix} f_{00} & f_{01} & f_{02} \\ f_{10} & f_{11} & f_{12} \\ f_{20} & f_{21} & f_{22} \end{bmatrix} \cdot [1\ 1\ 0]^T \\ &= \begin{bmatrix} f_{00} + f_{01} \\ f_{10} + f_{11} \\ f_{20} + f_{21} \end{bmatrix} = \begin{bmatrix} \rho_r P_{r00} + \rho_t P_t + \rho_b P_b + \rho_r P_{r01} \\ \rho_r P_{r10} + \rho_r P_{r11} \\ \rho_t P_{t20} + \rho_t P_{t21} \end{bmatrix} \end{aligned}. \tag{23}$$

Then, its simplified expression for DOLP is as follows:

$$\begin{aligned} \text{DOLP} &= \sqrt{\left((f_{10} + f_{11})^2 + (f_{20} + f_{21})^2\right)} / (f_{00} + f_{01}) \\ &= \sqrt{\rho_r^2 \cdot (P_{r10} + P_{r11})^2 + \rho_t^2 (P_{t20} + P_{t21})^2} / (\rho_r P_{r00} + \rho_t P_t + \rho_b P_b + \rho_r P_{r01}) \end{aligned}. \tag{24}$$

By writing Equation (25) in Mueller's form, the expression of Equation (24) becomes the following:

$$\begin{aligned} \text{DOLP} &= (\rho_r \times \frac{(M_{10} + M_{11})^r f_{SO}(\theta, \beta, \tau, \Omega) P_{RHG}(\cos\theta)}{4\cos\theta_i \cos\theta_r} \\ &+ \rho_t \times \frac{(M_{20} + M_{21})^t f_{tSO}(\theta_t, \beta_t, \tau, \Omega) P_{THG}(\cos\theta_t)}{4\cos\theta_r}) / f_{pBTDF} \end{aligned}. \tag{25}$$

By substituting the elements of the Müller matrix into Equation (25), the expression for the DOLP part of the pBTDF model is obtained as follows:

$$\begin{aligned} \text{DOLP} &= (\rho_r \times \frac{M_{ij}{}^r f_{SO}(\theta, \beta, \tau, \Omega) P_{RHG}(\cos\theta)}{4\cos\theta_i \cos\theta_r} \cos(2\psi) \\ &+ \rho_e \times \frac{M_{ij}{}^e f_{eSO}(\theta_e, \beta_e, \tau, \Omega) P_{EHG}(\cos\theta_e)}{4\cos\theta_r} \sin(\frac{2\beta}{\eta})) / f_{pBEDF} \end{aligned}. \tag{26}$$

Here, $\eta$ is the scale factor of the polarisation.

### 2.7. Experimental Verification

Equation (26) indicates that the DOLP is a multivariate nonlinear correlation function between the complex refractive index, reflected scattering component, and detection angle. The inversion of the five parameters, $n$, $k$, $\rho_r$, $\rho_t$, and $\rho_b$, of the target was performed using nonlinear least squares. An objective function was established, and the minimum standard deviation of the experimental measurements and model simulations was used as the best criterion for determining the model parameters [20], using the following expression:

$$\min\Delta E(n, k, \rho_r, \rho_t, \rho_b) = \frac{\sum_{\theta_i} \sum_{\theta_r} [\text{DOP}(\theta_i, \theta_r, \Delta\varphi) - \text{DOP}_m(\theta_i, \theta_r, \Delta\varphi)]^2}{\sum_{\theta_i} \sum_{\theta_r} [\text{DOP}_m(\theta_i, \theta_r, \Delta\varphi)]^2}, \tag{27}$$

where $\text{DOP}(\theta_i, \theta_r, \Delta\varphi)$ is the simulated value of the model, and $\text{DOP}_m(\theta_i, \theta_r, \Delta\varphi)$ is the experimental value obtained from the measurement.

To obtain the actual measurement values, an experimental scheme for evaluating the polarisation characteristics of indoor target surfaces was designed, and an experimental setup was built to obtain the polarisation images of the targets of different materials at different angles. The experimental system for polarisation characteristic testing included three parts: an active illumination device, a polarisation characteristic testing device, and

an information-processing device. A schematic of the experimental setup is shown in Figure 6.

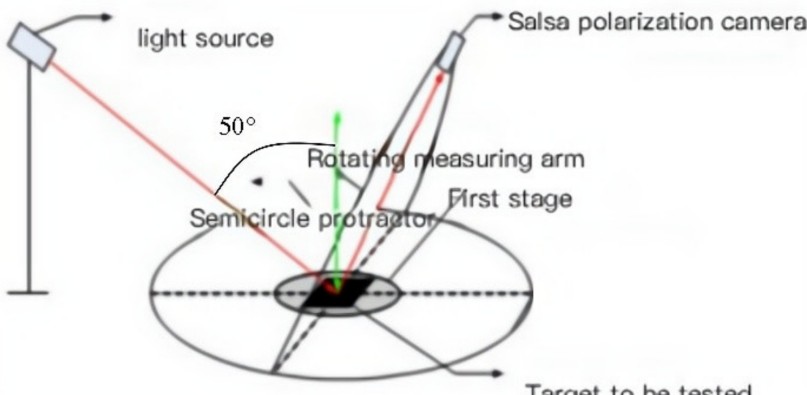

**Figure 6.** Diagram of the experimental setup.

The polarised light active illumination unit consists of a tripod, light source, polariser, and filters. The light source and polariser were placed on a tripod and aligned along the same optical axis. The polarisation characteristic test device was a SALSA polarisation camera, which controlled the detection angle and distance. The information-processing device consisted of a controller and a computer. The general idea of the experiment is that the light source passes through the polariser and becomes fully linearly polarised and then hits the target surface, where scattering occurs. The polarisation camera receives the information in real time and passes it to the controller, where the data are saved and calculated using a laptop.

The controller connects the information acquisition device to the information-processing device and has interfaces on the sides of the camera and computer. The information-processing device can synthesise images acquired using a polarisation camera to obtain the desired polarisation images and calculate the DOLP data. In the experimental verification, the polarisation image acquired using the above method and the calculated DOLP were used as the actual measurement data. A schematic of the DOLP acquisition process is presented in Figure 7 [12].

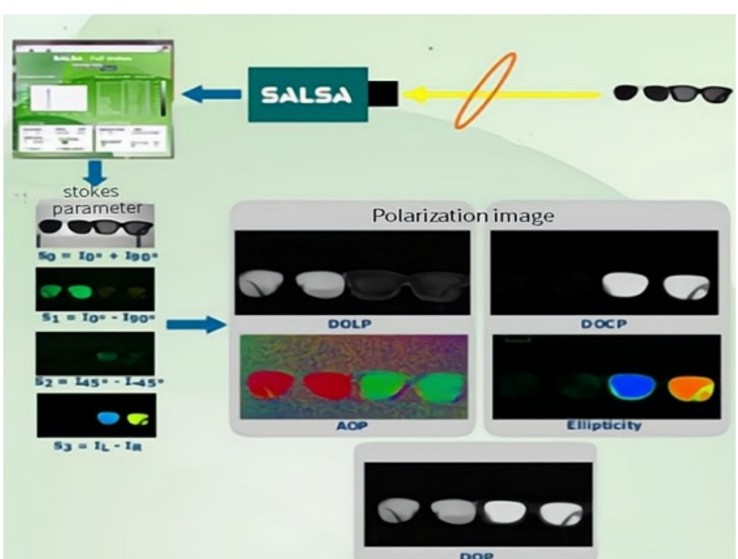

**Figure 7.** Degree of linear polarisation (DOLP) schematic.

Because the improved model based on diffraction and transmission effects is based on the special properties of common artefacts such as paint, those effects are modelled.

To verify the accuracy of the improved model, this experiment chose an aluminium plate as the substrate, and the coatings were chosen from light green, dark green, and earth yellow paints with a thickness of 0.2 mm, which visible light cannot penetrate; so, it can be considered the polarisation property of the paint surface. Different light wavelengths affect the DOLP [21]. Therefore, to avoid the effect of different wavelengths of light on the experimental results, we added a green filter in front of the SALSA camera lens with a central wavelength of 532 nm, allowing only green light waves to pass through and limiting the light's wavelength range. Polarisation images were acquired as follows:

Step 1: We placed the test sample in the centre and rotated the polariser to 0° so that the light source received 0° line-polarised light through the polariser.

Step 2: We set the zenith angle of detection normal to the target surface to 0°. Using this normal as a boundary, we set the detection zenith angle on the same side as the light-source emission direction to negative, and then we set the detection zenith angle on the different side to positive. We rotated the rotation arm on the zenith two-dimensional turntable to the −90° scale and increased the zenith angle by 2° each time until it moved from the −90° to 90° semicircle to complete the corresponding polarisation image acquisition.

Step 3: We added filters with different centre bands in front of the light source and repeated Steps 1 and 2 to acquire the polarisation information of the polarised image.

Step 4: We synthesised the captured polarisation images, using the controller to obtain the DOLP images and determine the corresponding DOLP as the actual measurement data of the DOLP.

The built experimental setup and data-acquisition site are shown in Figure 8.

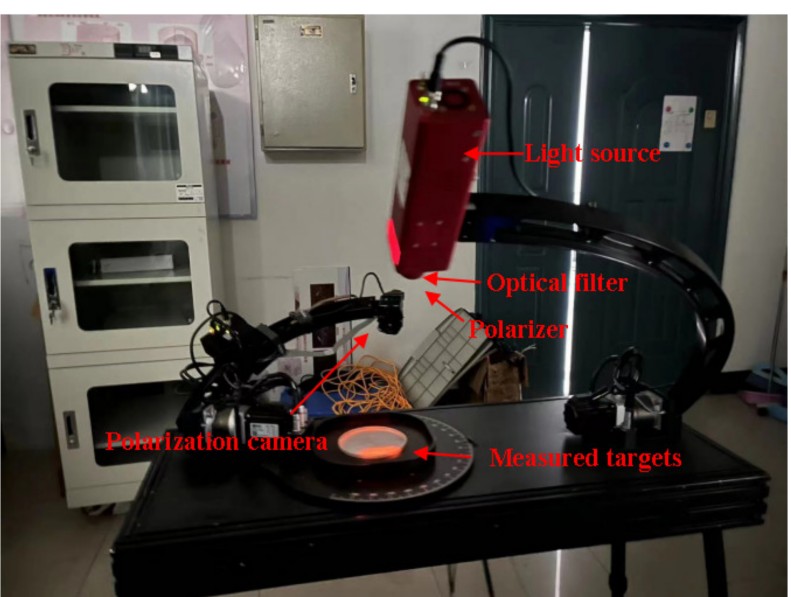

**Figure 8.** Data-acquisition site map.

### 3. Results

*3.1. Target Surface Polarisation Characterisation*

By substituting the measured and simulated data into Equation (27), the inverse results of the model parameters were obtained. The above inversion parameters were substituted into Equations (21) and (26) and the P-G model. The improved and P-G models were applied under the condition that the incident direction was coplanar with the observation direction. The variation in the DOLP and intensity with the detection zenith angle on the surface of the three coated aluminium plates were simulated, and the simulation results were plotted as curves and compared with the measured data, as shown in Figure 9.

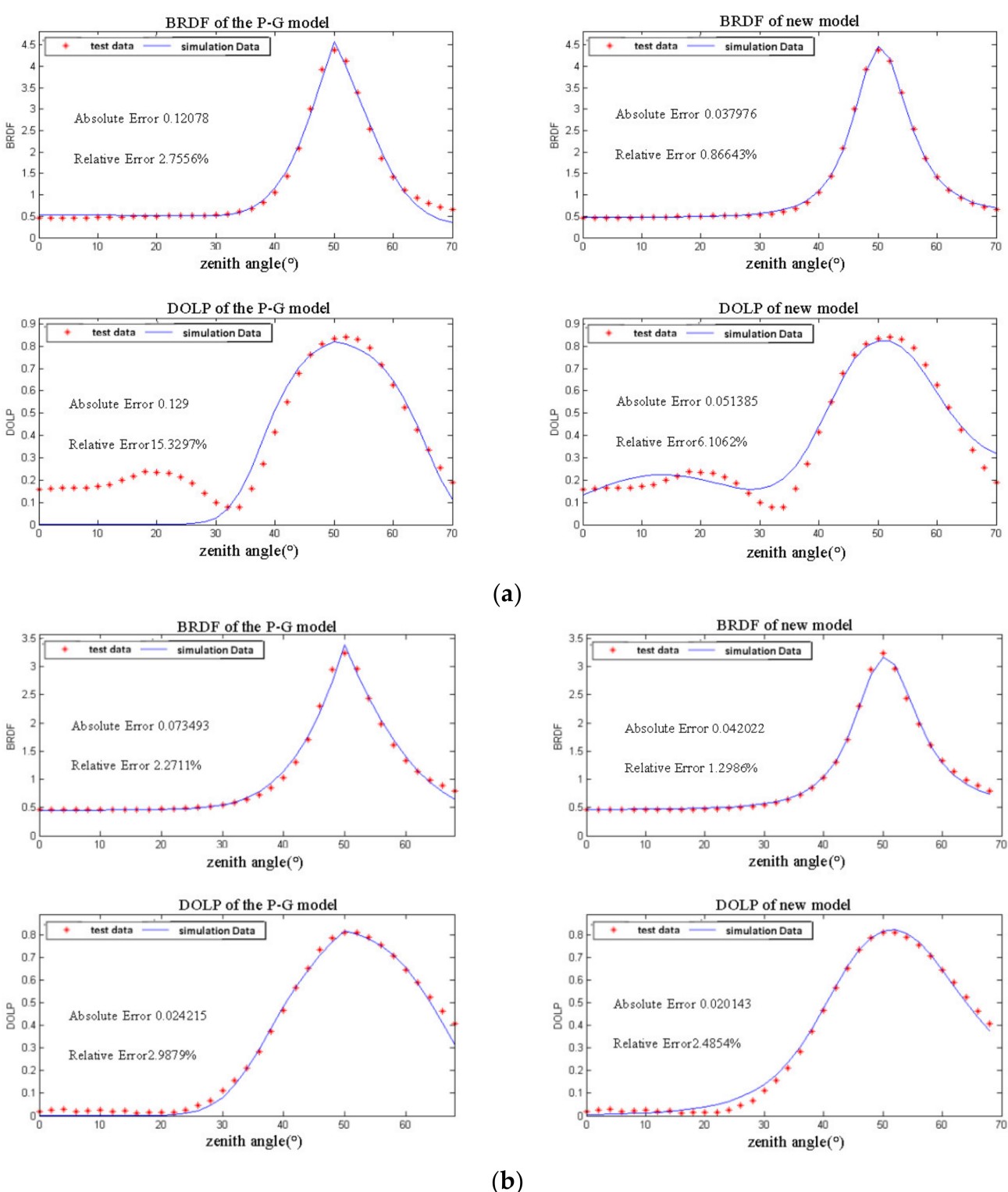

**Figure 9.** *Cont.*

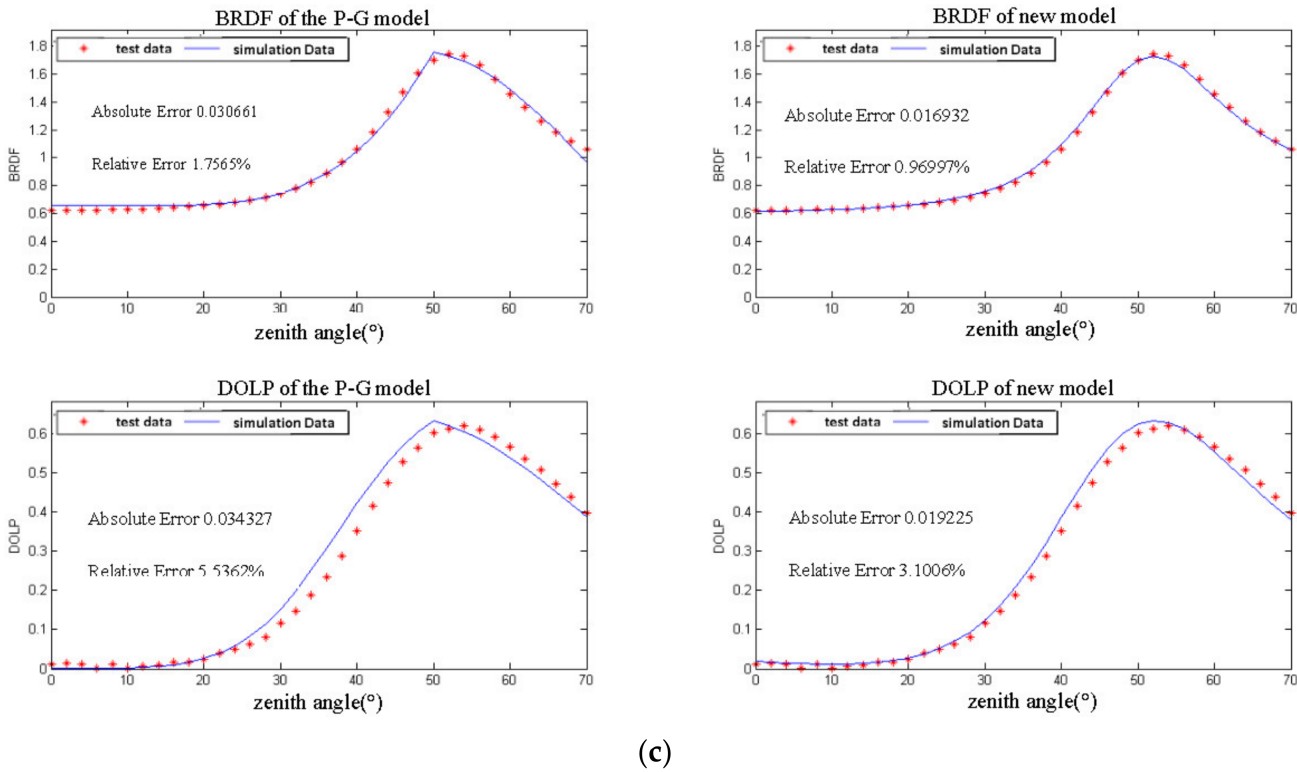

**Figure 9.** Comparative images of intensity and DOLP for aluminium plates under two models. (**a**) Intensity and DOLP comparison images of light green lacquer-coated aluminium panels under two models. (**b**) Intensity and DOLP comparison images of dark green lacquer-coated aluminium panels under two models. (**c**) Intensity and DOLP comparison images of earth yellow lacquer-coated aluminium panels under two models.

As can be seen from Figure 9, the DOLP of the three target materials shows a trend of becoming larger first and then decreasing with an increase in the detection zenith angle; furthermore, the peak occurs near a detection zenith angle of 50°, which is the direction of specular reflection of the incident light. This is because the specular reflection concentrates the reflected light in one direction, and so the intensity of linearly polarised light per unit area is the highest, and the DOLP is also the highest. Therefore, the greater the proportion of specular reflection, the greater the proportion of light received by the detector and the intensity of the polarised light, and the higher the DOLP will be. The transmission effect is a phenomenon that causes the light to rotate relative to the original specular reflection direction because of the presence of a curved or raised surface pattern of the target. Since all three targets are coated materials, their surfaces are relatively uniform, and the proportion of transmission is lower than that of specular reflection. The diffraction effect is because of a random change in the phase relationship between photons and photons, and this random change in phase leads to a decrease in the DOLP. Therefore, the DOLP of the target material depends mainly on the proportion of specular reflection that occurs. As the detection zenith angle increases, the proportion of specular reflection gradually increases, and when the detection zenith angle increases to 50°, the detector receives most of the light from the specular reflection in the direction of the specular reflection of the light source; thus, the intensity of the linearly polarised light is the highest, and the DOLP is also the highest. Therefore, the DOLP of the three target materials shows a trend of first becoming larger and then decreasing with the increase of the detection zenith angle. The angle located at the specular reflection of the incident light can be selected for polarisation detection of the target, which can improve the detection accuracy and identification capability.

### 3.2. Model Accuracy Verification

The proposed pBTDF model has better fitting accuracy than the P-G model, both in terms of intensity and DOLP. In the data processing, we used the root-mean-square error between the simulated and measured data of the model as the absolute error, intensity as the ratio of the root mean square error value to the maximum intensity value, and DOLP as the ratio of the root-mean-square error value to the maximum DOLP as the relative error.

The decline rate of the relative error was determined by calculating the ratio of the difference between the relative error of the P-G model and the improved model to the relative error of the P-G model and was $(2.7556 - 0.86643)/2.7556 = 0.6856 = 68.56\%$ for the intensity model of the light green painted aluminium plate, as an example. We used this value as a measure of the rate of decline of the relative error of the model and to judge the accuracy of the model.

As can be seen from Table 1, the relative errors of the three targets under the improved model were reduced by 68.56%, 42.80%, and 44.79%, respectively, compared to the traditional P-G model. The relative errors of the DOLP were reduced by 60.17%, 16.83%, and 44.00%, respectively. The experimental results and data validate that the proposed pBTDF model has better accuracy.

In order to analyse the accuracy of the proposed model more critically, this paper is compared with other improved models. Reference model 1 [22] gives an improved model for coating materials; however, the fitting accuracy is significantly lower than the improved model in this paper. Table 2 gives the root mean square error of reference model 2 [23], reference model 3 [20], and reference model 4 [10], and the results of comparison with the model in this paper.

As can be seen from Table 2, the root-mean-square error of the improved model proposed in this paper decreases by 49.07% compared with reference model 2, 48.43% compared with reference model 3, and 32.89% compared with reference model 4. It can be seen that the model in this paper has high accuracy for coating materials.

**Table 1.** Percentage decrease in intensity and DOLP relative error for the three targets under the two models.

| | Light Green Lacquer-Coated Aluminium Panels | | Dark Green Lacquer-Coated Aluminium Panels | | Earth Yellow Lacquer-Coated Aluminium Panels | |
|---|---|---|---|---|---|---|
| | **Relative Error (%)** | **Decline Rate (%)** | **Relative Error (%)** | **Decline Rate (%%)** | **Relative Error (%)** | **Decline Rate (%%)** |
| P-G model—intensity | 2.756 | ~ | 2.271 | ~ | 1.757 | ~ |
| pBTDF model—intensity | 0.866 | 68.56 | 1.299 | 42.80 | 0.97 | 44.79 |
| P-G model—DOLP | 15.33 | ~ | 2.988 | ~ | 5.536 | ~ |
| pBTDF model—DOLP | 6.106 | 60.17 | 2.485 | 16.83 | 3.100 | 44.00 |

**Table 2.** The DOLP root-mean-square error of the model in this paper and the reference model.

| | **Model of this Paper** | **Reference Model 2** | **Decline Rate (%)** |
|---|---|---|---|
| DOLP root mean square error of light green lacquer-coated aluminium panels | 6.1062% | 11.991% | 49.07% |
| | **Model of This Paper** | **Reference Model 3** | **Decline Rate (%)** |
| DOLP root mean square error of dark green lacquer-coated aluminium panels | 2.4854% | 4.82% | 48.43% |
| | **Model of This Paper** | **Reference Model 4** | **Decline Rate (%)** |
| DOLP root mean square error of earth yellow lacquer-coated aluminium panels | 3.1006% | 4.62% | 32.89% |

## 4. Discussion

The study of polarisation characteristics of target surfaces is of great importance in the fields of polarisation detection and polarisation remote sensing. Using the target polarisation characteristic model, the accuracy of polarisation detection can be improved, and it has high application value in scenarios such as military stealth target detection, atmospheric transportation, and computer vision. Compared with the traditional P-G model, both the intensity and the accuracy of the proposed model of DOLP are improved by a large percentage. In practical applications, the model has higher sensitivity to coated materials or materials with strong light transmission, which can provide better discrimination and identification capabilities and a good basis for material detection. The model also provides rigorous theoretical support for the study of target surface polarisation characteristics and can be applied to the software system of target surface polarisation characteristics measurement devices such as ellipsometers.

The proposed pBTDF model can provide a more accurate theoretical basis for the study of target surface polarisation properties, but it still has limitations and potential challenges. For some strong scatterer materials, the use of a simple single spherical particle scattering superposition does not answer the question of multiple scattering in strong scatterers, and the modelling of multiple scattering effects is still needed for this type of material. Second, the specular reflection part can be accurately represented by Fresnel's formula, but for the non-specular reflection part, that is, the part that cannot be accurately described by Fresnel's formula, only the primary reflection of the specular micro-element is considered; there is no description of the multiple reflection effect. Therefore, it is necessary to model the multiple reflection effect in subsequent research to further improve the accuracy of the model.

## 5. Conclusions

A pBTDF model was proposed and established based on the improvement of the traditional pBRDF model, using transmission and diffraction effects. The least-squares method was used to invert the target parameters, the inversion results were substituted into the model, and the simulation curves and measurements that were obtained had a better fitting effect. By comparing the relative errors of intensity and DOLP between the improved model and the conventional P-G model, it was concluded that the relative errors of intensity for the three targets were reduced by 68.56%, 42.80%, and 44.79%, respectively, under the improved model. The relative errors of the DOLP were reduced by 60.17%, 16.83%, and 44.00%, respectively. This proves that the proposed pBTDF model has improved accuracy. The DOLP values of the samples varied at different detection angles. The DOLP of the target was highest in the specular reflection direction of the beam. This suggests that the specular reflection direction is the ideal detection angle for polarisation detection instruments, and this is in line with conventional knowledge.

In this study, the internal intrinsic properties affecting the polarisation characteristics of a target were investigated through a theoretical derivation to obtain a more accurate analytical expression for the pBRDF model. This can provide a basis for the selection of pBRDF models for different processes, such as target identification, matter detection, and atmospheric transport. This can contribute to future research and applications of target polarisation properties.

**Author Contributions:** Data curation, writing—review and editing, conceptualisation, and methodology, Q.F.; data curation, writing—review and editing, and formal analysis, X.L.; visualisation and investigation, D.Y.; investigation, J.Z.; resources, Q.L.; resources, S.Z.; software, F.W.; software, J.D.; writing–review and editing, Y.L.; supervision, H.J. All authors have read and agreed to the published version of the manuscript.

**Funding:** This research was funded by the National Natural Science Foundation of China (grant numbers 61890960, 61890965, and 61890963).

**Institutional Review Board Statement:** Not applicable.

**Informed Consent Statement:** Not applicable.

**Data Availability Statement:** Data underlying the results presented in this paper are not publicly available at this time but may be obtained from the authors upon reasonable request.

**Conflicts of Interest:** The authors declare no conflict of interest.

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
