# Peer review of "Improvement of pBRDF Model for Target Surface Based on Diffraction and Transmission Effects"

_remotesensing, doi:10.3390/rs15143481_

Round 1
Reviewer 1 Report
This paper tried to improve the pBRDF model of the target surface based on diffraction and transmission effects. However, this manuscript is written unclearly. Especially, why the diffraction and transmission effects are important for the pBRDF model? And, in which conditions should consider the influence of diffraction and transmission effects? It should be firstly and emphatically analyzed. Since this is the core point of view and innovation of this manuscript.
In general, the writing of the paper appears sloppy, with many obvious errors, hence I cannot recommend it for publication in the journal of Remote Sensing at current stage. My specific comments for this paper are as follows:
1. What type of surface is the pBRDF model established in this manuscript suitable for? Smooth surface or rough surface? It should be specified.
2. Since the P-G model is the first full pBRDF model, many improved pBRDF based on the P-G model have been proposed in the following twenty years. However, the introduction of relevant research is missing in the manuscript. The results of this manuscript only give the relative error between the improved model and the conventional P-G model, but do not show the simulation difference from other improved pBRDF models. So, it’s hard to measure the contribution of this work.
3. How was the data in Fig. 3 obtained? Which method was used? And what does the vertical axis of Fig. 3 represent?
4. Fig. 4 seems to be incompletely drawn, and it is not clear what it means.
5. Fig. 5(b) and Fig. 5(c) are the same picture.
6. Fig. 6 does not indicate what the different lines represent.
7. Fig. 8 and Fig. 9 have already appeared in other publications, hence, the corresponding citation should be added.
Author Response
Dear reviewer,
We are very grateful for your comments on the manuscript. This is a response letter about “Improvement of pBRDF model for target surface based on diffraction and transmission effects”. We have revised the original manuscript according to your kind advice and detailed suggestions. The manuscript is resubmitted, and we wish it could be reconsidered for publication in “Remote Sensing”.
Below is the response to the Reviewer’s advice and detailed suggestions. The reviewers’ comments and our replies are listed, respectively. The corrections are red marked and underlined in the revised manuscript for easy check purposes.
If there are other needs, please feel free to inform us.
Thank you very much!
Best regards!
Yours sincerely,
First author: Qiang Fu
Corresponding author: Qiang Fu

Reviewer 2 Report
In this paper, the authors claim to improve the conventional P-G model (based on a combination of specular microelements, which need to have a scale much larger than the wavelength) with 40% for both diffraction and transmission effects. The experimental results validate that the proposed pBTDF model has better correctness and accuracy.
Please explain why Figure 5 b) similar with Figure 5 c).
Legend could be added in Figure 6 for a better understanding.
The results presented in this paper could have various applications such as atmospheric transport, matter detection, target identification.
Author Response

(The authors gave the same response as above.)

Reviewer 3 Report
The authors study the bidirectional reflectance distribution function for the three different lacquers on the aluminum substrate. The main idea is to improve the functionality of this polarization-based method using the modified diffraction- and transmission-related phase contributions. I cannot recommend this manuscript to be published in Remote Sensing owing to the following reasons.
1) I consider this contribution as the incremental one. The model improvements include the basics of optical physics. So, I do not see enough novelty and originality in this manuscript. The authors should specify the uniqueness of this research and its impact on the optical community.
2) The authors concluded that the relative errors of intensity and degree of linear polarization (DOLP) are reduced by 45-68% and 23-48%, respectively, with their modified new model. It is an ambiguous point and even a bit of data manipulation. One can see from Fig. 10 that the standard and modified models give pretty the same results (except one special case of the DOLP for the earth yellow lacquer), especially in the vicinity of the peaks at the spectra. It seems to me that the relative error in each point does not exceed 10%, but the authors calculate the integral error (sum of the errors in each point) as far as I understand. So, it gives the incorrect picture of the results obtained.
3) The manuscript is not well-organized. It is quite difficult for reading, many details are omitted, the figures are of a bad quality. The manuscript contains a lot of misprints (for instance, “nm” instead of “um” in the line 130, “Meuller” instead of “Mueller” in the line 189, etc.)
To conclude, I recommend to consider this manuscript for this or more specific journal only after a substantial revision.
English may be improved. The misprints must be eliminated.
Author Response

(The authors gave the same response as above.)

Reviewer 4 Report
The manuscript proposes a pBTDF model that enhances the traditional pBRDF model by incorporating transmission and diffraction effects. Using the least-squares method, the model demonstrates improved fitting and reduced relative errors of intensity and DOLP. However, further clarification is needed in certain sections, and a more comprehensive analysis of the statistical significance and limitations of the proposed model is required.
The following points need to address:
1. The manuscript lacks a thorough discussion of the limitations and potential challenges of the proposed pBTDF model.
2. The statistical significance of the observed reduction in relative intensity errors and DOLP should be further analyzed and presented.
3. The clarity of the presentation could be improved in certain sections, particularly in explaining the theoretical derivation of the internal intrinsic properties affecting the polarization characteristics.
4. Authors should write shorter sentences. As in the case of abstract “In this paper…..”.
5. Introduction must be written in a clearer way.
6. Lots of typographical mistakes are preset for, eg. In lines 47 and 114, “the photons and the photons,”. In line no 119, the end of the sentence is missing.
7. What is the significance of E in equation 6, authors must include the term's relevance in the text and how it affects the study.
8. Overall, authors must proofread the complete manuscript to avoid grammatical and typographical mistakes.
Moderate editing of English language required.
Author Response

(The authors gave the same response as above.)

Round 2
Reviewer 1 Report
The authors have addressed the issues of my concern.
Reviewer 3 Report
The authors have referred to all my comments and improved the manuscript. I suggest it for the publication in Remote Sensing journal. However, the manuscript still contains some misprints. I also suggest to improve the quality of Figs. 6, 7 and 9.
The manuscript still contains some misprints.